# Active Surveillance for Prostate Cancer: Past, Current, and Future Trends

**DOI:** 10.3390/jpm13040629

**Published:** 2023-04-03

**Authors:** Ivo I. de Vos, Henk B. Luiting, Monique J. Roobol

**Affiliations:** Department of Urology, Erasmus MC Cancer Institute, University Medical Center Rotterdam, 3015 GD Rotterdam, The Netherlandsm.roobol@erasmusmc.nl (M.J.R.)

**Keywords:** prostatic neoplasms, prostate cancer, active surveillance, expectant management, review

## Abstract

In response to the rising incidence of indolent, low-risk prostate cancer (PCa) due to increased prostate-specific antigen (PSA) screening in the 1990s, active surveillance (AS) emerged as a treatment modality to combat overtreatment by delaying or avoiding unnecessary definitive treatment and its associated morbidity. AS consists of regular monitoring of PSA levels, digital rectal exams, medical imaging, and prostate biopsies, so that definitive treatment is only offered when deemed necessary. This paper provides a narrative review of the evolution of AS since its inception and an overview of its current landscape and challenges. Although AS was initially only performed in a study setting, numerous studies have provided evidence for the safety and efficacy of AS which has led guidelines to recommend it as a treatment option for patients with low-risk PCa. For intermediate-risk disease, AS appears to be a viable option for those with favourable clinical characteristics. Over the years, the inclusion criteria, follow-up schedule and triggers for definitive treatment have evolved based on the results of various large AS cohorts. Given the burdensome nature of repeat biopsies, risk-based dynamic monitoring may further reduce overtreatment by avoiding repeat biopsies in selected patients.

## 1. Introduction

Currently, prostate cancer (PCa) is the most diagnosed malignancy in men in the most industrialised countries and the second leading cause of male cancer death worldwide [1]. With the growth and ageing of the population, the worldwide number of PCa cases is expected to grow to almost 2.4 million new cases per year by 2040. Before the introduction of prostate-specific antigen (PSA) testing [2], as many as one in five patients diagnosed with PCa had advanced disease at diagnosis [3,4,5]. In the early 1990s, the widespread adoption of PSA testing caused an increase in the incidence of PCa and a significant shift from metastatic to localised disease at the time of diagnosis [4,6]. This revolutionised early detection of PCa as level 1 evidence demonstrated that population-based PSA screening leads to a substantial reduction in PCa-specific mortality, as well as a reduction in the incidence of metastatic disease [7,8]. Despite this evidence, the indolent course of many screen-detected tumours (i.e., cancer that would otherwise not become clinically manifest over a patient’s lifetime or not result in cancer-related death) generated controversy regarding the utility of screening for PCa [9]. Definitive treatment with surgery or radiotherapy of these indolent tumours often coincides with harmful side effects that can severely impact quality of life, including urinary incontinence, sexual dysfunction, and bowel dysfunction [10]. Consequently, active surveillance (AS) was introduced as an alternative for immediate definitive treatment for low-risk PCa in order to counteract the harm of so-called overdiagnosis and subsequent overtreatment. Rather than offering immediate definitive treatment, AS is an approach that uses a combination of PSA testing, digital rectal examinations (DRE), imaging, and prostate biopsies to monitor PCa. Unlike watchful waiting where PCa patients receive palliative treatment in case of symptoms, AS rests on the assumption that curative treatment within the window of cure is still possible if deemed necessary. This allows definitive treatment and its possible side effects to be postponed or even omitted altogether if there are no signs of progression, thus reducing overtreatment.

The paper aims to review the evolution of AS over the past 25 years and to provide an overview of its current state and challenges.

## 2. History and Establishment of Active Surveillance Studies

The first paper that coined the term “active surveillance” was published in 1990 [11]. Smith pointed out the long natural history of low-grade, localised PCa and that immediate treatment may not always be of benefit since the majority of these patients will not die of PCa. Instead, he proposed to monitor men with well-differentiated PCa and to only offer treatment to those patients in whom progression was observed. However, it was not yet clear which specific men should be offered AS and how to monitor these men. Based on the principle of reducing overtreatment while still retaining a window of cure, several questions about AS needed clarification. First, for which men is AS a safe alternative to radical therapy? Second, how do we need to monitor men on AS? Finally, what should be a trigger for switching to a definitive treatment? Several studies from the 1990s provided evidence for determining the suitable candidates for AS. Long-term observational data from the pre-PSA era showed that the PCa-specific survival at 10–15 years after diagnosis among patients with localised PCa who were treated conservatively (i.e., watchful waiting) was strongly related to tumour grade and patient comorbidities [12,13,14,15,16]. In a pooled analysis of 828 conservatively treated patients conducted by Chodak et al., the 10-year PCa-specific survival rate for grade 1 (i.e., Gleason score 2–4) and grade 2 (i.e., Gleason score 5–7) tumours was 87% [12]. Additionally, for grade 1 and grade 2 tumours with clinical stage T1a, the 10-year disease-specific survival rate was found to be even higher: 96% and 94%, respectively. Another study by Johansson et al. revealed that the 15-year PCa-specific survival of 81% was similar between patients with localised disease (T0–T2) with deferred treatment and initial treatment [13]. These studies demonstrated the long natural history of PCa and provided supporting evidence that patients with localised disease and a life expectancy of fewer than 10 years could be managed expectantly with watchful waiting. On the other hand, it also revealed that men with low-grade, localised tumours and a life expectancy of more than 10 years could be feasible candidates for AS to improve survival rates even more while still reducing overtreatment. Furthermore, in 1994, Epstein et al. proposed a set of criteria based on PSA and biopsy features which identified potentially biologically insignificant tumours that might be safely managed by initial surveillance [17]. Considering the risk of prostate biopsies for undersampling, leading to underestimating tumour grade and amount, the authors compared biopsy results with radical prostatectomy pathology in non-palpable tumour patients. Based on this comparison they developed the following criteria predicting insignificant tumours on needle biopsy: clinical stage T1c, PSA density <0.15 ng/mL, no Gleason pattern 4 or 5, <3 positive cores, and <50% cancer per core. Another risk classification system was developed by D’Amico et al. in 1998 [18]. This staging system which is based on PSA, clinical T-stage, and Gleason score, stratified patients into low-, intermediate-, or high-risk of biochemical recurrence (BCR) of disease after radical prostatectomy or radiotherapy.

To establish the safety, feasibility, and approach of AS, prospective studies with predefined clinical parameters and a specified follow-up protocol were initiated. It was not until 2002 that the first results of prospective protocol-managed AS cohorts were published by the University of Toronto [19] and the Johns Hopkins Medical Institute [20]. During the first decade of 2000, additional prospective AS cohorts from other institutions followed: Memorial Sloan Kettering Cancer Centre (MSKCC [21]); Prostate Cancer Research International: Active Surveillance study (PRIAS [22]); Royal Marsden Hospital (RMH [23]); University of California San Francisco (UCSF [24]); and the Canary Prostate Active Surveillance Study (Canary PASS [25]). These studies all employed a protocol-driven AS strategy to monitor patients with a low-grade, localised disease where selective delayed intervention was indicated based on clinical progression, PSA kinetics and/or histological progression on repeat biopsy. 

The eligibility criteria used in these trials were largely similar to the Epstein criteria and the low-risk group by D’Amico. When these studies were initiated, the degree of restriction in inclusion criteria varied (see Table 1). Varying combinations of Gleason score, PSA level, and clinical T-stage were applied. Four cohorts were limited to the inclusion of Gleason score 3 + 3 disease, while three cohorts also allowed 3 + 4 tumours. Most studies had a PSA level threshold of <10 ng/mL for inclusion, however, some studies had other thresholds, such as the Toronto study allowing PSA levels up to 15 ng/mL for men aged >70, and Canary PASS having no PSA threshold. All cohorts only included localised disease (<T2), but there was variation in which subgroup (a–c) of T2-stage was allowed and the Johns Hopkins cohort limited inclusion to patients with non-palpable tumours. Other criteria to consider were PSA density and tumour volume characteristics based on percent positive cores or the extent of cancer in any core. 

The follow-up schedules contained serial PSA measurements, DRE, and re-biopsies to check for signs of tumour progression. Most cohorts monitored men with PSA testing at intervals of 3–12 months, DRE at intervals of 6–12 months, a confirmatory biopsy within a time frame of one year, and follow-up biopsy at intervals of 1–3 years. There especially was consensus concerning the importance of an early confirmatory biopsy to minimise the risk of biopsy undersampling. 

The criteria for converting to active treatment were determined by disease progression, with three categories being considered: clinical progression, PSA kinetics, and histological progression. Clinical progression was usually defined as an increase in T-stage on digital rectal examination. PSA kinetics, such as a PSA-DT of less than 3 years or an absolute increase of >0.75/1 ng/mL per year, were used in most studies as a surrogate for tumour progression and therefore trigger intervention. This was based on limited evidence that an increase in pre-treatment PSA level was associated with adverse pathology at radical prostatectomy [26] or even biochemical recurrence following definitive treatment [27]. Furthermore, histological progressions on re-biopsies, such as an upgrade in Gleason score or increased tumour volume, served as a trigger for intervention.

## 3. Evidence for Active Surveillance from Randomised Controlled Trials Comparing Definitive Treatment and Observation

Although the results of the first two AS studies in 2002 indicated the feasibility of an AS protocol, the short-term outcome data presented was restricted to the progression rate to treatment and were considered preliminary results [19,20]. Longer-term outcomes regarding the safety of AS, such as metastasis or PCa-specific mortality, were much anticipated. Meanwhile, results from three randomised controlled trials involving different populations of men, which compared the efficacy of immediate definitive treatment with watchful waiting or active monitoring of localised PCa, emphasised the potential value of AS. The publication of the SPCG-4 trial results in 2011 [28], the PIVOT trial results in 2012 [29], and the ProtecT trial results in 2016 [30] provided evidence of the long natural history of PCa and which patients might benefit most from AS. 

The SPCG-4 trial enrolled 695 men with localised PCa between 1989 and 1999 (pre-PSA era). Participants were randomised to undergo radical prostatectomy or watchful waiting. The mean age of participants was 65 years, with a mean PSA level of 13 ng/mL, and 24% of tumours were clinical stage ≤T1c. After a median follow-up of 12.8 years, there was a significant relative reduction in overall mortality (25%) and PCa-specific mortality (38%) in favour of radical prostatectomy. However, subset analysis of the study demonstrated very low numbers of PCa-specific mortality in men with low-risk disease (PSA < 10 ng/mL and Gleason score <7) and men older than 65 years with no significant difference between the two arms. This was also highlighted in a post hoc analysis which showed that younger patients with high-grade and advanced clinical stage PCa had the most benefit from radical prostatectomy [31]. 

The PIVOT trial randomised 731 men with localised PCa to either radical prostatectomy or watchful waiting between 1994 and 2002 (early era of PSA testing). The mean age of participants was 67 years, with a mean PSA level of 7.8 ng/mL. Most tumours (54%) were clinical stage ≤T1c. After a median follow-up period of 10 years, no significant difference in overall survival or PCa-specific mortality was observed between the two treatment arms, except amongst men with PSA > 10 ng/mL. Of the 148 men D’Amico low-risk disease in the watchful waiting arm, only 4 men died of their disease which was not significantly different from the 6 of 148 low-risk disease men in the radical prostatectomy arm.

In contrast to SPCG-4 and PIVOT, the ProtecT trial monitored men for disease progression using protocoled PSA testing. Between 1999 and 2009, 1643 men with screen-detected localised PCa with a median age of 62 years and median PSA 4.6 ng/mL were randomised to radical prostatectomy, radical radiotherapy, or active monitoring. Although the majority (76%) of men had clinical stage ≤ T1c, at least 28% of the men had intermediate- or high-risk disease according to contemporary risk-stratification [32]. Monitoring consisted of PSA testing every 3 months in the first year and every 6–12 months thereafter without protocoled re-biopsies. If PSA levels rose more than 50% in a year, the patient was considered for definitive treatment. After a median follow-up of 10 years, the primary intention-to-treat analysis showed a rate of overall mortality of 10% and a rate of PCSM of 1.0%, with no significant difference between the treatment arms. Although there was a significantly higher rate of metastasis in the active monitoring arm than in the definitive treatment arm, this was thought to be driven by the intermediate- and high-risk PCa’s in the monitoring arm. Analysis of the association of baseline characteristics with disease progression confirmed this notion [33]. Furthermore, despite the anticipation that this higher rate of metastatic disease in the active monitoring arm at 10 years post-diagnosis would affect prostate cancer-specific mortality (PCSM) in the long run, the latest update from the ProtecT trial again revealed no significant difference in PCSM after a median follow-up of 15 years [32].

The effect of urinary, bowel, and sexual function on the quality of life was also assessed in the ProtecT study [34,35]. In the monitoring group, sexual erectile function, as well as urinary continence and function, were less affected compared to the definitive treatment groups, but gradually declined over time as men became older and more men received definitive treatment in the monitoring arm.

Findings from these randomised controlled trials emphasised the validity of offering watchful waiting for men with a life expectancy of fewer than 10 years and provided evidence that although definitive treatment is probably best for younger men with intermediate- to high-risk disease, AS should be a viable option for men with low-risk PCa and a life expectancy greater than 10 years who are often detected via opportunistic PSA screening.

## 4. Evolution of Active Surveillance Inclusion Criteria and Intervention Triggers

As intermediate- and long-term results from AS cohorts showed favourable results in terms of low rates of metastases and PCa-specific deaths, confidence in AS as a treatment strategy for low-risk PCa grew. In 2015, a systematic review by Simpkin et al. revealed only 8 PCa deaths and 5 cases of metastases in 26 AS cohorts consisting of 7627 men and 24,981 person-years of follow-up [36]. However, uncertainties regarding optimal patient selection and reliable intervention criteria persisted since different AS strategies resulted in varying rates of change to definitive treatment across studies, from 1.1% to 22% per year [36]. The effect of inclusion and intervention criteria on intermediate outcomes such as progression on rebiopsy, adverse pathology after radical prostatectomy, and biochemical recurrence after definitive treatment were assessed. In addition, with the introduction of multiparametric magnetic resonance imaging (mpMRI) in AS, these findings prompted adjustments in the inclusion and intervention criteria (Table 1). 

First, the use of PSA kinetics in AS protocols as a trigger for immediate intervention became debated. An analysis from the Johns Hopkins group showed that PSA kinetics did not reliably predict histopathological progression on rebiopsy [37]. They demonstrated that PSA doubling time and PSA velocity were not significantly associated with subsequent adverse biopsy findings. In addition, there was no PSA velocity or PSA doubling time threshold that had both high sensitivity and specificity for progression on biopsy. Other studies from UCSF, Royal Marsden Hospital, and the University of Miami also supported the notion that PSA kinetics did not provide adequate prognostic value to justify the use as a sole indicator for intervention in AS programs [38,39,40]. Furthermore, data from the PRIAS cohort showed that even though men with a PSA double time of fewer than 3 years had a twice higher risk of upgrading in Gleason score on rebiopsy, 46% of those undergoing radical prostatectomy due to fast-rising PSA had favourable pathology (i.e., Gleason 3 + 3 and pT2) [41,42]. As a result, most major AS programs now consider PSA kinetics as a reason for further diagnostic evaluation rather than as a trigger to initiate definitive treatment (Table 1).

In addition, the implementation of mpMRI in the diagnostic pathway of PCa had implications for AS. mpMRI in combination with MRI-targeted biopsy showed improved detection rates of clinically significant cancer and reduced rates of clinically insignificant cancer in comparison with standard systematic biopsy alone [43]. With the introduction of these targeted biopsies, utilisation of tumour volume as a criterion for inclusion and intervention became questionable. While tumour volume characteristics on systematic biopsy such as the number of positive cores or the extent of cancer in any core were shown to be significant predictors of progression on rebiopsy [44], a targeted biopsy could cause inflation to the tumour volume and may therefore overestimate that risk. Additionally, long-term data showed that Gleason 6 tumours treated with radical prostatectomy did not develop metastases nor die from PCa irrespective of tumour volume at diagnosis [45,46]. Therefore, tumour volume on systematic biopsy was mainly thought to be a surrogate marker for higher-risk disease, and targeted biopsy could reduce or even eliminate this issue of undersampling. Consequently, most AS-managed cohorts dropped tumour volume of Gleason 6 disease as an inclusion criterion and as a trigger for intervention when mpMRI was used at inclusion and/or in follow-up (Table 1).

Moreover, with growing support for AS in low-risk disease and validated safety through numerous studies, there had been growing interest in expanding its application to patients with intermediate-risk PCa. The introduction of mpMRI could improve the initial selection for AS by reducing the undersampling of higher-risk tumours. Nevertheless, the use of MRI-targeted biopsy could also result in stage migration. For instance, a patient classified as having low-risk PCa based on systematic biopsy could be reclassified as intermediate-risk based on the finding of a low amount of Gleason 4 on MRI-targeted biopsy. In a study by Ahdoot et al. where MRI-targeted and systematic biopsies were compared, this stage migration occurred in 20% of the men who had Gleason 3 + 3 on systematic biopsy [47]. However, it was uncertain if MRI-detected cancers pose the same long-term oncologic risk as those detected by systematic biopsy with the same grade. Kovac et al. found that patients diagnosed with low-grade disease on biopsy but with high-grade cancer on surgical pathology did not have substantially higher rates of recurrence and mortality compared to low-risk patients without upstaging [48]. This raised questions about whether the inclusion criteria of AS should be corrected for this stage shift. The only long-term data of men with intermediate PCa on AS was from the Toronto cohort which included men with Gleason 3 + 4 tumours early on, which reflects AS before the introduction of mpMRI. In their report in 2016, they demonstrated that the 15-year metastasis-free survival, overall survival, and cancer-specific survival were all worse in the intermediate-risk group than in the low-risk group [49]. However, at 10 years post-diagnosis the metastasis-free survival rate of 98% for men with Gleason 3 + 4 and PSA ≤ 20 ng/mL was comparable to the 95.3% rate for men with low-risk PCa. Despite a decline in metastasis-free survival to 83% after 15 years, this was not statistically different from the survival rate of men with low-risk PCa. This suggested that AS may be a viable option for selected patients with Gleason 3 + 4 PCa. This was also supported by an updated report from the PIVOT trial with a median follow-up of 18.6 years that showed no significant benefit in overall survival between surgery and observation for men with Gleason 7 at diagnosis [50]. Due to the heterogeneity of this risk group, the challenge is to identify the men with intermediate-risk tumours who may have the same indolent course as low-risk men. A study by Kweldam et al. found that Gleason 3 + 4 tumours have varying clinical behaviour, depending on subtypes of Gleason 4 growth patterns [51]. Cribriform growth was associated with unfavourable outcomes and metastasis, while Gleason 3 + 4 without cribriform growth demonstrated similar clinical behaviour to Gleason 3 + 3. The authors suggested that patients with PCa who have a Gleason 3 + 4 score without cribriform growth could be suitable candidates for AS. Overall, these data suggested that AS for intermediate-risk PCa carried some degree of risk but could be acceptable to a subset of patients with favourable characteristics in combination with the use of mpMRI. This led AS protocols to expand their inclusion to selected men with Gleason 3 + 4 as they also implemented the use of mpMRI in their inclusion and follow-up protocol (Table 1). Results from a recent meta-analysis of intermediate-risk patients on AS also support the inclusion of patients with low-volume Gleason 3 + 4 tumours as oncologic outcomes of these patients appeared similar to those with low-risk PCa [52].

## 5. Current Guideline Recommendations and Uptake of AS

In the last years, prospective protocol-managed AS cohorts have published their long-term results [42,53,54,55,56,57]. These long-term outcomes show good 10- to 15-year metastasis-free and PCa-specific survival rates ranging from 95–100% for low-risk PCa. In a pooled analysis of the Global Action Plan Prostate Cancer Active Surveillance (GAP3), Bruinsma et al. combined data from 25 centres across 15 countries and included information on 15,101 men from various AS cohorts, of which over 1000 patients have at least 10 years follow-up [58]. In these data, only 45 men (0.3%) developed metastases, and 37 (0.2%) died from PCa. After 5, 10, and 15 years of follow-up, 58%, 39%, and 23% were still on AS, while 23%, 30%, and 36% discontinued AS due to progression based on protocols.

The publication of long-term results was instrumental in converting AS from its previous status as an investigational approach to a standard of care for low-risk disease in most international guidelines. Table 2 shows the current recommendations of widely used guidelines: EAU, AUA, NICE, and NCCN [59,60,61,62]. Guidelines use PSA, ISUP grade group (i.e., the updated Gleason grading system [63]; [GG]), clinical T-stage, PSA density, and tumour volume for patient selection. The stratification of newly diagnosed PCa in these guidelines is more or less based on the original D’Amico risk classification [18], but the definitions of favourable intermediate-risk (GG2) vary between the guidelines. The general consensus is that AS is the preferred treatment for men with a life expectancy of >10 years and low-risk PCa (GG1 with PSA < 10 ng/mL and ≤T2a), and an optional treatment for favourable intermediate-risk (GG1 with PSA < 20 ng/mL and ≤T2a, or, i.e., GG1 with PSA < 10 ng/mL and ≤T2a with low tumour volume).

As confidence in the long-term safety of AS has grown, the uptake of AS in men with low-risk disease has increased. In the United States, low-risk patients who are initially treated with AS increased from 27% in 2014 to 60% in 2021 [64]. In the Netherlands, 85% of the newly diagnosed men with very low-risk PCa in 2015–2016 were managed with AS [65]. In a Swedish study that covered 98% of newly diagnosed PC cases from 2009 to 2014, the usage of AS increased from 40% to 74% for low-risk and from 57% to 91% for very low-risk PC cases, which shows that AS is now the dominant treatment in these men [66]. The uptake of AS for men with intermediate-risk disease is lower. In Sweden, this number increased from 31% in 2009 to 53% in 2014 for men with GG1 and PSA 10–20 ng/mL. In contrast, the uptake for men with GG2 and a PSA <20 ng/mL only slightly increased from 14% in 2009 to 17% in 2014 [66]. 

## 6. Barriers to Uptake and Compliance of Active Surveillance

Although a positive trend is observed in the adoption of AS among eligible patients for surveillance, the numbers are still not ideal. There are several barriers regarding the uptake of AS, which may be patient- or clinician-related. A comprehensive systematic review of factors affecting both choice and adherence to AS identified multiple barriers and facilitators, including patient characteristics such as age and cancer features, social and family support, attitudes conveyed by healthcare providers, and influences from healthcare organisations and policies [67]. Based on these factors, the authors made several suggestions on how to improve the uptake of AS, such as harmonising national/local guidelines, improving shared decision-making, and improving patient education and information, but also raising awareness through social media. Another review by Cunningham et al. in which they explored men’s perspectives on the factors that influence their decision-making process when considering AS, emphasised the need for individualised, clear, and relevant information to support men in making informed choices [68]. The authors indicated that clinicians should explore personal factors, such as a man’s (familial) cancer-related experiences and perception of risk and discuss them in relation to their specific cancer diagnosis.

Moreover, compliance with the follow-up protocol or discontinuing AS without following protocol guidance remains a challenge. Compliance is particularly poor regarding repeat biopsies [41,69]. Prostate biopsies can be invasive, uncomfortable and not without risk of infection or significant bleeding [70]. Bokhorst et al. showed that according to data from more than 4000 men in the PRIAS study, only approximately 30% of them underwent all repeat biopsies specified by the protocol [41]. In particular, yearly biopsies because of faster rising PSA were often ignored, even though these men were at higher risk of upgrading on repeat biopsies. PRIAS data also showed that men with a prior biopsy complication, such as infection, haematuria, haematospermia, or pain, are less likely to undergo a repeat biopsy as scheduled [71]. On top of that, 66–90% of repeat biopsies may be considered redundant as they do not show histological progression [72,73,74], which is now the main trigger for discontinuing AS. The burden and fear of biopsies can also lead the patient to discontinue AS. Data from the GAP3 showed that approximately 13% of the men who discontinued AS did so without evidence of disease progression [75]. Reducing the frequency of biopsies when it is generally safe to do so, could reduce the burden of AS and improve compliance with biopsies.

## 7. Risk-Based Follow-Up in Active Surveillance

Personalised risk-based follow-up may avoid unnecessary biopsies and treatment, while still minimising the delay of the detection of progression. Current AS approaches usually utilise a one-size-fits-all approach with fixed, frequent biopsies, but this does not consider individual progression rates. Fast-progressing patients benefit from frequent biopsies, but slow-progressing patients are subjected to unnecessary, burdensome biopsies. Hence, a trade-off should be made on an individual basis between the burden of biopsy and the time delay in detection of upgrading. Consensus exists for a confirmatory biopsy after 1 year, but heterogeneity remains for subsequent repeat biopsies. Personalised, dynamic, risk calculators using individual clinical data may empower clinicians and patients to better understand the risk and make informed decisions about repeat biopsies. This approach was also underlined as the most important research priority by a recent expert consensus meeting [76]. Several prediction models based on clinical characteristics have been developed already [77,78,79,80,81,82,83,84]. The development cohorts, statistical technique, included clinical variables, and outcomes vary between these models (Table 3). The Johns Hopkins and PRIAS models employ a dynamic approach by utilising repeated measurements, whereas the Canary PASS and STRATCANS models rely on data from diagnosis or the latest follow-up visit. While these models provide promising results, head-to-head comparison and feasibility assessment in an AS risk-based protocol are required before implementing these models in daily clinical practice.

In addition to the conventional clinical characteristics such as PSA-kinetics and DRE, imaging with serial prostate MRI has become prevalent in AS follow-up protocols which may aid in personalised risk-based follow-up. While the STRATCANS model incorporates MRI findings at diagnosis, it does not account for radiological changes during follow-up. Studies indicate that radiological changes on MRI during AS are predictive of histological progression, potentially reducing unnecessary biopsies [85,86,87]. Nonetheless, a recent systematic review showed that serial MRI alone still lacks sufficient accuracy and should only be used in conjunction with other clinical biomarkers, such as PSA and PSA density [88]. 

Furthermore, as various PCa-related genetic mutations have been identified and genetic risk assessment is becoming more common in the diagnosis of PCa, genetic testing may also play a role in the risk stratification and selection for AS [89]. These genetic factors can be combined into individual genetic risk scores which allows for a more accurate assessment of the patient. Even though early studies show that these scores are associated with reclassification and discontinuation of AS [90,91], the cost-effectiveness of such genetic risk scores remains unknown.

## 8. Conclusions

Over the last 25 years, AS has evolved and is now a standard of care strategy in the management of low-risk PCa and can also be considered in selected patients with favourable intermediate-risk disease. Although the uptake of AS has increased over the years, barriers to the uptake and compliance of AS remain. As patient selection is improved and personalised, dynamically adaptive follow-up becomes available, and men may require less invasive monitoring in the future so that overtreatment can be reduced even further.

## Figures and Tables

**Table 1 jpm-13-00629-t001:** Protocol-driven, prospective active surveillance cohorts.

Institution	Start Study	Inclusion Criteria				Follow-Up Schedule	Criteria Triggering Definitive Treatment	Evolution of Protocol
		Gleason Score	PSA (ng/mL)/PSA Density(ng/mL^2^)	Tumour Stage	Tumour Volume			
University of Toronto	1995	3 + 3 and 3 + 4 if aged >70 years	≤10 or ≤15 and >70 years old/NR	≤T2b	NR	PSA every 3 months for 2 years then every 6 months Biopsy within 6–12 months, then every 3–4 years	Clinical progression based on DRE or urinary symptoms Histopathological features: any upgrading in Gleason score PSA kinetics: PSA-DT <3 years (2 years until 1999)	Inclusion: restricted to men with 3 + 3 and PSA ≤ 10 ng/mL or men with PSA 10–20 and/or 3+4 with significant comorbidities and a life expectancy <10 years Follow-up: adverse PSA kinetics triggers MRI Intervention criteria: PSA kinetics was discontinued as a trigger for intervention; Clinical progression triggers biopsy instead immediate active treatment
Johns Hopkins Medical Institute	1995	3 + 3	NR/ ≤ 0.15	≤T1c	≤2 positive cores, and <50% cancer per core	PSA/DRE every 6 months Biopsy yearly	Histopathological features: ≥3 + 4; 3 positive cores; >50% cancer per core	Inclusion: expanded to men with 3 + 3, ≤T2a, and a PSA < 10 ng/mL Follow-up: MRI included (interval not specified) Intervention criteria: increased tumour volume was discontinued as a trigger for intervention
Memorial Sloan Kettering Cancer Centre	2000	3 + 3	≤10/NR	≤T2a	≤2 positive cores, and ≤50% cancer per core	PSA/DRE every 3 months for 1 year, then every 6 months Biopsy yearly or if PSA/DRE/TRUS showed progression	≥3 score based on histopathological features and PSA kinetics	Inclusion: expanded to 3 + 3 with no limitation on PSA level or number of positive cores. ≤3 + 4 and/or ≤T2b are also allowed Follow-up: PSA/DRE every 6 months; MRI every 18 months; confirmatory biopsy within 12 months and biopsy every 2–3 years or in case of MRI/PSA progression Intervention criteria: PSA kinetics and increased tumour volume were discontinued as a trigger for intervention
PRIAS	2006	3 + 3	≤10/≤0.2	≤T2c	≤2 positive cores	PSA every 3 months for 2 years then every 6 months Biopsy at year 1,4 and 7 Yearly biopsies if PSA-DT between 3–10 years	Clinical progression to ≥T3 Histopathological features: ≥3 + 4; ≥3 positive cores PSA kinetics: PSA-DT <3 years	Inclusion: expanded to higher PSA (≤20), PSA density (≤0.25) and no limit in the number of positive cores when MRI is used at inclusion; Gleason 3 + 4 without cribriform/intraductal carcinoma with ≤50% cores positive is also allowed Follow-up: PSADT < 10 years triggers yearly MRI; DRE only yearly after 2 years Intervention criteria: PSA kinetics and increased tumour volume were discontinued as a trigger for intervention
Royal Marsden Hospital	2002	3 + 3 and 3 + 4 if aged >65 years	≤15/NR	≤T2a	≤10 mm cancer of any core, and <50% positive cores	PSA/DRE every 3 months for 2 years, then every 6 months Biopsy at 1 year, then every 3 years	Histopathological features: ≥4 + 3; 50% cores positive PSA kinetics: increase of >1.0 ng/mL per year	Inclusion: MRI for all patients at inclusion Follow-up: MRI every 2 years
University of California San Francisco	1990	3 + 3	≤10/NR	≤T2a	<33% positive cores	PSA/DRE every 3 months TRUS every 6–12 months Starting 2003, repeat biopsies every 12–24 months	Histopathological features: ≥3 + 4 PSA kinetics: increase of >0.75 ng/mL per year	Inclusion: men who do not meet the criteria can enrol in the study after shared decision-making Follow-up: biopsy within 12 months; interval MRI Intervention criteria: PSA kinetics was discontinued as a trigger for intervention
Canary Prostate Active Surveillance Study	2008	3 + 3 and 3 + 4	No limitations	≤T2c	NR	PSA every 3 months DRE every 6 months Biopsy within 6–12 months, at 2 years, then every 2 years	Clinical progression based on DRE Histopathological features: any upgrading in Gleason score PSA kinetics: PSA-DT <3 year	No changes

NR: not reported; PSA: prostate-specific antigen; DRE: digital rectal examination; TRUS; transrectal ultrasound; PSA-DT: prostate-specific antigen double-time; MRI: magnetic resonance imaging.

**Table 2 jpm-13-00629-t002:** Current guideline recommendations on active surveillance.

Guidelines	ISUP Grade Group	PSA (ng/mL)	Clinical Tumour Stage	PSA Density (ng/mL/g)	Tumour Volume	Strength of Evidence	Other Recommendations
EAU	1	<10	≤T2a	NR	NR	Strong	Life expectancy should be >10 y Perform MRI in AS patients who have not had an MRI previously Exclude patients with intraductal and cribriform histology
	2	<10	≤T2a	NR	<10% pattern 4; ≤3 cores positive; and ≤50% core involvement/per core	Weak	
AUA	1	<20	≤T2a	NR	NR	Strong	Life expectancy must be taken into account
	2	<10	≤T2a	“low”	“Low” % of pattern 4; and <50% of total cores positive	Strong	
NICE	1	<20	≤T2	NR	NR	NR	Perform MRI in AS patients who have not had an MRI previously
	2	<10	≤T2	NR	NR	NR	
NCCN	1	<20	≤T2a	≤0.15	NR	NR	Life expectancy should be >10 y
	2	<10	≤T2a	“low”	“Low” % of pattern 4; and <50% of total cores positive	NR	

EAU: European Association of Urology; AUA: American Association of Urology; NICE: National Institute for Health and Care Excellence; NCCN: National Comprehensive Cancer Network; ISUP: International Society of Urological Pathology; NR: not reported; PSA: prostate-specific antigen; MRI: magnetic resonance imaging; AS: Active Surveillance.

**Table 3 jpm-13-00629-t003:** Prediction models for reclassification in active surveillance.

Prediction Model	Development Cohort	Statistical Technique	Included Clinical Variables	Outcome	Performance	External Validation
Johns Hopkins [77,78]	964 patients Gleason score ≤6 and at least two PSA measurements and at least 1 post-diagnosis biopsy	Dynamic Bayesian joint model	Repeated PSA and biopsy results	Gleason score ≥3 + 4 at radical prostatectomy	AUC = 0.74 (95%CI: 0.66–0.80)	None
Canary Prostate Active Surveillance Study [79,80]	859 patients with Gleason score ≤6 at least 1 post-diagnosis biopsy	Logistic regression with generalised estimating equations	Most recent PSA; PSA change; age; time since the most recent prior biopsy; negative biopsy after biopsy; and the percent of positive cores (<34% vs. ≥34%) on the most recent prior biopsy	Gleason score ≥3 + 4 or an increase in percentage of cancer cores positive to ≥34% upon repeat biopsy	AUC = 0.72	Johns Hopkins: AUC = 0.75 MSKCC: AUC = 0.68 PRIAS: AUC = 0.63 Toronto: AUC = 0.69 UCSF: AUC = 0.67
Canary Prostate Active Surveillance Study [81]	850 patients with Gleason score ≤6 and at least 1 post-diagnosis biopsy	Partly conditional Cox proportional hazards regression	PSA and prostate volume at diagnosis; PSA-kinetics; time since diagnosis; negative biopsy after diagnosis; maximum percent positive cores at diagnosis; and body mass index	No reclassification at 4 years	AUC = 0.70 (95%CI: 0.63–0.76)	UCSF: AUC = 0.70
PRIAS [82,83]	7813 patients with Gleason score ≤6	Dynamic Bayesian joint model	Repeated PSA and biopsy results; timing of prior biopsy; and age at inclusion	Gleason score ≥3 + 4 upon repeat biopsy	Time-dependent AUC = 0.62–0.69	Johns Hopkins: AUC = 0.60–0.74 MSKCC: AUC = 0.58–0.75 Toronto: AUC = 0.64–0.79 UCSF: AUC = 0.62–0.74 KCL: AUC = 0.68–0.69 MUSIC: AUC = 0.60
STRATCANS Model [84]	883 patients with Gleason score ≤3 + 4	Cox proportional hazards regression	At diagnosis: PSA; Gleason score; prostate volume; percent of positive cores; MRI PI-RADS score; age; and family history	Gleason score ≥4 + 3 or Gleason score ≥3 + 4 with PSA ≥ 10 upon repeat biopsy	C-index = 0.74 (95%CI: 0.69–0.79)	Cardiff: C-index = 0.85

PSA: prostate-specific antigen; MRI: magnetic resonance imaging; PI-RADS: Prostate Imaging Reporting and Data System; AUC: area under the curve; MSKCC: Memorial Sloan Kettering Cancer Centre; UCSF: University of California San Francisco; KCL: King’s College London; MUSIC: Michigan Urological Surgery Improvement Collaborative.

## Data Availability

Data sharing not applicable.

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
