# Peer review of "Active Surveillance for Prostate Cancer: Past, Current, and Future Trends"

_jpm, 2023, doi:10.3390/jpm13040629_

Round 1

Reviewer 1 Report

The aim of the article by de Vos et al., was to provide a comprehensive review of the history of active surveillance and its current impact for prostate cancer management. In particular, the authors analyze the literature and nicely describe the rationale behind the development of active surveillance, a type of watchful waiting and monitoring that allows for reduction of overtreatment and improvement of prostate cancer patient quality of life. This is a well-written review that offers a comprehensive description of the current active surveillance status and could be of interest to a broad spectrum of journal readers. However, the last part of this review referring to the “future trends” of active surveillance is somewhat lacking and would benefit from expansion. See comments below:

·      The authors touch upon the risk assessment approach for active surveillance but do not go into details on how and why this could actually improve prostate cancer prognosis. There have been comprehensive reviews on this matter (PMID: 36609003, 35490919 36551580) that the authors should consider including and utilizing to further expand on how the future of active surveillance may progress.

·      The discussion on the use of multiparametric (mp)MRI in the context of active surveillance is a vet important point the authors nicely point out but should be expanded as to how this could also be used in future applications, maybe in conjugation with genetic testing, further improve active surveillance in prostate cancer.

Author Response

The aim of the article by de Vos et al., was to provide a comprehensive review of the history of active surveillance and its current impact for prostate cancer management. In particular, the authors analyze the literature and nicely describe the rationale behind the development of active surveillance, a type of watchful waiting and monitoring that allows for reduction of overtreatment and improvement of prostate cancer patient quality of life. This is a well-written review that offers a comprehensive description of the current active surveillance status and could be of interest to a broad spectrum of journal readers. However, the last part of this review referring to the “future trends” of active surveillance is somewhat lacking and would benefit from expansion. See comments below:

  • The authors touch upon the risk assessment approach for active surveillance but do not go into details on how and why this could actually improve prostate cancer prognosis. There have been comprehensive reviews on this matter (PMID: 36609003, 35490919 36551580) that the authors should consider including and utilizing to further expand on how the future of active surveillance may progress.

Response 1: We would like to thank the reviewer for the valuable peer-review of our manuscript. We have revised the future trends by also including the impact of genetic testing in active surveillance.

  • The discussion on the use of multiparametric (mp)MRI in the context of active surveillance is a vet important point the authors nicely point out but should be expanded as to how this could also be used in future applications, maybe in conjugation with genetic testing, further improve active surveillance in prostate cancer.

Response 2: We have further expanded the future trends of the use of MRI in active surveillance.

Reviewer 2 Report

By implementing active surveillance, unnecessary treatment can be avoided, as well as extreme side effects, for men with clinically localized prostate cancer who have a life expectancy of more than 10 years, and ensuring that the correct timing is achieved for curative treatment in those who ultimately require it, even if they do not require immediate treatment.

This method is presented in a narrative form to demonstrate how it has gained a place within specialized medical guidelines, the current viewpoint for current practice, and potential future applications.

It is well-written, contains many references, and touches on the main issues that have arisen on this topic over the years.

For this review to comply with the recommendations of the journal, I suggest that the authors include an Introduction section and extend the Future Directions in a separate subsection.

There is also a significant number of self-citations from some coauthors of the manuscript.

Once the manuscript has undergone modifications and an overall English verification, it will be suitable for publication.

Author Response

By implementing active surveillance, unnecessary treatment can be avoided, as well as extreme side effects, for men with clinically localized prostate cancer who have a life expectancy of more than 10 years, and ensuring that the correct timing is achieved for curative treatment in those who ultimately require it, even if they do not require immediate treatment.

This method is presented in a narrative form to demonstrate how it has gained a place within specialized medical guidelines, the current viewpoint for current practice, and potential future applications.

It is well-written, contains many references, and touches on the main issues that have arisen on this topic over the years.

For this review to comply with the recommendations of the journal, I suggest that the authors include an Introduction section and extend the Future Directions in a separate subsection.

Response 1: We would like to thank the reviewer for the critical appraisal of our manuscript. We have revised our manuscript accordingly to comply with the recommendations of the journal. We have changed the first section to the introduction section. In addition we extended the future perspectives.

There is also a significant number of self-citations from some coauthors of the manuscript.

Response 2: We understand your concern, however, we believe that the number of self-citations is inevitable in this case. One of the co-authors has been actively researching and publishing in the field of active surveillance for many years, resulting in a significant body of work that is highly relevant to the current study. Additionally, the field of active surveillance is highly collaborative, and the co-authors have been involved in many collaborations (e.g. GAP3) resulting in publications that are widely cited in the field and have contributed significantly to our understanding of active surveillance. Therefore, we believe that these citations are essential to the overall review of the literature in this area.

Once the manuscript has undergone modifications and an overall English verification, it will be suitable for publication.

Round 2

Reviewer 2 Report

A manuscript revision has been made and the manuscript is suitable for publication.